# A Survey from Real-Time to Near Real-Time Applications in Fog Computing Environments

**Eliza Gomes** [1,2,*], **Felipe Costa** [1], **Carlos De Rolt** [3], **Patricia Plentz** [1] and **Mario Dantas** [4]

1. Department of Informatics and Statistics, Federal University of Santa Catarina, Florianopolis 88040-900, SC, Brazil; felipe.costa@ifsc.edu.br (F.C.); patricia.plentz@ufsc.br (P.P.)
2. BRy Tecnologia, Florianopolis 88036-002, SC, Brazil
3. Centre of Management and Socioeconomic Science, State University of Santa Catarina, Florianopolis 88035-001, SC, Brazil; rolt@udesc.br
4. Department of Computer Science, Federal University of Juiz de Fora, Juiz de Fora 36036-900, MG, Brazil; mario.dantas@ice.ufjf.br
* Correspondence: eliza.gomes@posgrad.ufsc.br

**Abstract:** In this article, we present a comprehensive survey on time-sensitive applications implemented in fog computing environments. The goal is to research what applications are being implemented in fog computing architectures and how the temporal requirements of these applications are being addressed. We also carried out a comprehensive analysis of the articles surveyed and separate them into categories, according to a pattern found in them. Our research is important for the area of real-time systems since the concept of systems that respond in real time has presented various understandings and concepts. This variability of concept has been due to the growing requirements for fast data communication and processing. Therefore, we present different concepts of real-time and near real-time systems found in the literature and currently accepted by the academic-scientific community. Finally, we conduct an analytical discussion of the characteristics and proposal of articles.

**Keywords:** real-time; near real-time; IoT; fog computing; edge computing

## 1. Introduction

The widespread use of devices connected to the Internet has spread the IoT approach and has changed the way people live and live with advanced technology. In a smart home, for example, the deployment of IoT devices aims to offer greater interoperability between home appliances. Health devices offer greater monitoring of elderly or disabled people and, consequently, greater independence for their activities. As a result, IoT has become an important topic for the technology industry and the academic area [1].

However, IoT applications require an environment with features not always present in cloud computing, such as support for mobility and geographic distribution, location recognition, low latency and fast response to assertive decision-making [2]. To meet these requirements, Bonomi et al. [3] proposes an intermediate platform called Fog Computing, which provides computing, storage, and networking services between edge devices and cloud computing data centers.

Most data generated by applications using IoT requires time-sensitive processing [4]. The requirements present in applications based on IoT, such as low latency and fast processing have been presented as essential characteristics of an application of temporal nature, and this has generated multiple definitions for systems that respond in real time. In this article, we present a survey on the use of the fog computing approach with time-sensitive applications. The goal is to research what applications are being implemented in fog computing architectures and how the temporal requirements of these applications are being addressed. We also carry out a bibliometrics analysis, so that it is possible to verify some characteristics present in the selected articles such as: countries and publishers

who generated more publications; the types of articles published; and the number of publications per year. Finally, we point out the lessons learned.

The main motivation for conducting this survey was to obtain the understanding presented by the community regarding the concept of real time and fog computing/edge computing. In addition, it seeks to analyze the most widespread applications for the evolution of future work.

The article is organized as follows: Section 2 presents the material and methods that consist of the concepts of edge computing, fog computing, and the difference between edge and fog computing, as well as the different concepts of real-time and near real-time presented in the literature. Moreover, we describe the research protocol proposed and used for the survey, and we analyze the selected articles and proposed a classification according to the applied real-time concept. Section 3 presents a summary of the articles according to the assigned category. In Section 4, we point out the lessons learned, carry out an analytical discussion about the selected articles, and present the indications of future works. Finally, Section 5 concludes the article and indicates some future works.

## 2. Material and Methods

In this section, we will present the material and methods used to carry out our research. In the context of this article, we consider material to describe the concepts and characteristics of the approaches and technologies that are the basis for our research. On the other hand, the research methodology details the methods used to search, select, and analyze the articles.

### 2.1. Material

In this section, we briefly introduce the concepts related to edge computing, fog computing, and real-time. These concepts are important, as they offer an overview and understanding of the systems, as well as how the environment that presents these approaches is structured. The goal is to provide a better understanding since it is possible to find in the literature different definitions on edge computing, fog computing, and real-time topics. Therefore, we present the difference between fog and edge computing, as well as three different concepts of real-time systems.

#### 2.1.1. Edge Computing

In the last few years, the scientific community needed to define new concepts to embrace the new computation models that increased IoT worldwide scenery. Edge concepts are in this set of new computational paradigms. As they are the basis for several new applications, it is worth noticing to understand them clearly.

Edge computing terminology does not have a single established definition. Currently, it is possible to find two biases that conceptualize edge and differentiate it, or not, from fog computing. (1) Edge computing is identified with users' devices and fog computing is an intermediate layer between edge and cloud computing; (2) Edge computing is considered a resource for communication, computing, control and storage, in close proximity to devices and end users. In this concept, there is no need to include the fog layer, since the edge and cloud layer perform all communication and processing [5].

From the point of view of this article, the edge refers to the devices, sensors, or other data sources at the edge of the network. In the center of the network, there is a data center, a cloud, which processes all data that arrives there. The management layer that arranges data between cloud and edge is the fog substrate [6].

Therefore, edge computing cannot be conceptualized as a data center or as a simple sensor that converts from analog to digital and collects/sends data [7]. It is possible to conceptualize edge computing as a layer composed of data providers (sensors) and mobile devices next to sensors/actuators with some processing capacity, such as smartphones, tablets, and PDAs [8].

The layer between the end devices and the cloud, also called the edge layer, could be implemented in different ways in terms of some variations: the devices which serve as the intermediate edge nodes, the networks, and communication protocols used by the edge layer as well as the services delivered by the edge layer [9]. Based on this context, the paper shows that three different implementations of Edge Computing can be employed: Fog Computing, Mobile Edge Computing and Cloudlet. The first one deals with leveraging M2M devices like gateways and wireless routers through Fog Computing Nodes. The second one, Mobile Edge Computing, suggests using cellular networks as base stations for intermediate nodes with storage and processing capabilities. In the last one, Cloudlet, the authors defend the use of dedicated devices with capacities similar to a data center but on a lower scale existing in consumers' surroundings.

Finally, it is essential to highlight that Fog computing is a distributed approach that comes down from its Edge computing nature and derives from the need to deal with the Cloud computing centralized approach [10].

### 2.1.2. Fog Computing

Fog Computing is a platform that provides processing, storage, and networking resources between edge devices and cloud computing data-centers [11]. Thus, fog computing can be seen as an extension that complements cloud computing and not as a replacement.

The main goal of fog computing is to support applications and services that are not serviced by cloud computing [10], which are:

- Applications that require predictable and low latency;
- Geographically distributed applications;
- Mobile applications that require fast responses;
- Large scale distributed control systems.

The three main layers (Edge Computing, Fog Computing, and Cloud Computing) of an IoT environment, presented in Figure 1, can be seen as a hierarchical organization of network resources, computing, and storage [12].

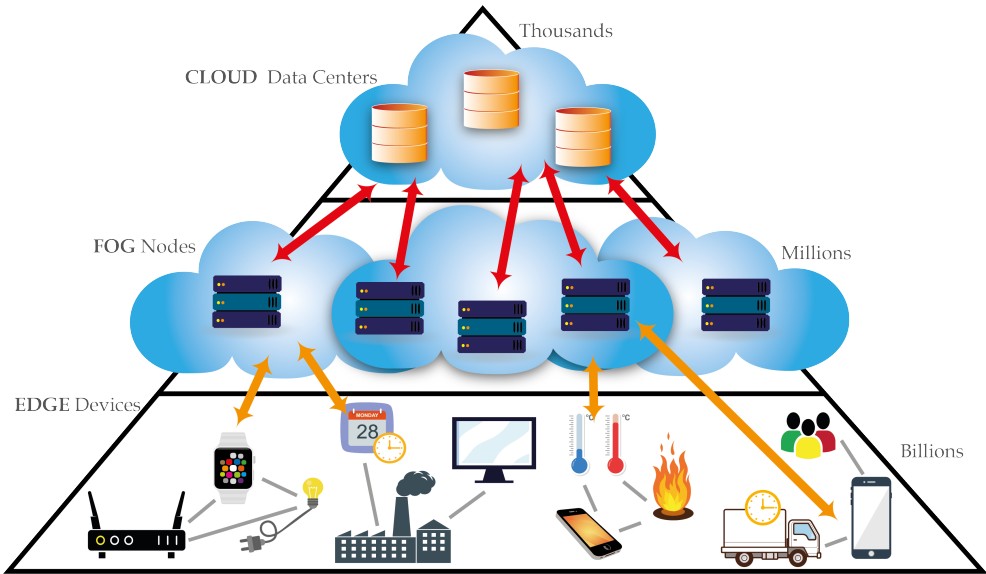

**Figure 1.** Hierarchical organization between edge, fog, and cloud computing.

According to Bonomi et al. [10], the interaction between the edge computing, fog computing, and cloud computing layers begins when the edge computing layer sends the data generated by the sensor and device grid to the fog computing layer. The first layer of fog, designed for interaction machine-to-machine (M2M), processes the data and sends commands to the actuators belonging to the Edge layer. This layer also filters the data to be used locally and sent to the upper layers. The second layer includes both system

processes and data delivery to the machine-to-machine cloud as well as human-to-machine interaction (H2M), which deals with visualization and reporting.

The time scale between these interactions are: from milliseconds to seconds between the edge and fog layer; seconds to minutes in the first and second layers of the fog; and minutes to days to send data to the cloud computing and transactional analysis [10].

Applications such as Smart Home, Smart Grid, Smart Vehicle, and Healthcare benefit from the advantages fog offers, namely low latency and fast response [12].

- Smart Home: the variety and heterogeneity of the devices and sensors connected in the house make connecting them difficult. In this case, fog can be used to integrate devices into a single platform and enable applications to handle the elasticity of resources;
- Smart Grid: it is an electricity distribution network, with smart meters deployed at various locations to measure state information instantly. A centralized server called the SCADA system gathers and analyzes state information and sends commands to respond to changing demand or emergencies to stabilize the power grid;
- Smart Vehicle: fog computing can be integrated into the vehicle network and can be categorized into two types: based on infrastructure and autonomous. The first is based on fog nodes deployed along the road; the nodes are responsible for sending/receiving information to/from the vehicle steering. The second uses moving vehicles to form the fog to support ad hoc events; each fog can communicate with clients and other fogs;
- Healthcare: health data are considered sensitive as it is valuable and private data. With fog computing, the storage and processing of this data are done locally, which makes it safer, faster to view, and allows the user to have their health information;

The temporal requirements of the applications may vary according to the characteristics they present. For example, smart vehicles and healthcare are applications with greater sensitivity to time, compared to smart home and smart grid, due to the immediate need for decision-making.

### 2.1.3. Differences Between Edge and Fog Computing

Fog computing is often referred to as edge computing, but, according to Iorga et al. [13], these two approaches have key differences, which are:

- Fog computing works with cloud computing while edge communicates only with the local network;
- Fog computing is hierarchically structured, while the edge has a limited number of peripheral layers;
- Unlike edge, the fog computing has support for network, storage, control, and acceleration of data processing.

What is Fog computing and how it differs from Edge computing are questions widespread among the scientific community mainly because these are new concepts in the consolidation process. The paper De Donno et al. [14] explores these questions and sheds light on their clarification. According to the authors, Fog computing is the highest evolution of the Edge computing principles. This strong affirmation is based on the idea that cloud computing could be extended to the network edge, near the data sources (IoT devices), to bring the computation inside these devices. This idea can be implemented in different ways, like Mobile Edge Computing, Mobile Cloud Computing, and Cloudlet Computing. However, in the authors' view, Fog Computing is not yet another implementation of Edge computing because it is not limited to only the edge of the network. It incorporates the Edge computing concept, offering support to a well-organized intermediate layer that fully bridges the gap between IoT and Cloud computing.

### 2.1.4. Real-Time and Near Real-Time

This subsection aims to present concepts of real-time and near real-time, which we understand to be the concept that applies most to unpredictable systems and with variable processing times, such as health monitoring.

The Gomes et al. [15] survey shows that the term "real-time" has been presented and accepted, in the academic area, with different concepts. The authors present three concepts of real-time respectively applied to digital real-time signal processing, real-time systems, and real-time data stream:

- Digital real-time signal processing: the hardware and software is designed to complete a task within a specified period of time.
- Real-time system: the system must present correct logical results, but it is imperative that it presents them within a deadline.
- Real-time data stream: the system must provide the processing high volumes of data, low latency, and fast response.

On the other hand, according to Kononenko et al. [16], Perera et al. [17], Osanaiye et al. [18], applications that run the data stream can be classified according to the response time as real-time or near real-time. For Perera et al. [17], near real-time can be understood as high response sensitivity, but without the hard time constraint. For Kononenko et al. [16], Osanaiye et al. [18], the data are immediately processed upon arrival. In other words, near real-time is almost the time of the live event if subtracting the current time from the processing time.

### 2.2. Research Methodology

In view of the different concepts of systems that respond in real-time found in the literature, we conducted an advanced search of articles that propose the use of real time and fog computing. The objective of this research is to verify how these two approaches have been used together and to analyze which bias the research has followed.

For this, we carry out a systematic literature review based on the procedures suggested by Kitchenham and Charters [19]. The procedures adopted were divided into three phases:

- Review Planning: it consists of the specification of the research questions and the validation of the search protocol;
- Conducting the Review: it consists of applying the search protocol and identifying and selecting articles;
- Analysis of Results: consists of the results obtained after reading and analyzing the proposals of the selected articles.

The following sections present the details of the procedures performed for this survey.

### 2.2.1. Review Planning

According to Kitchenham and Charters [19], this step is necessary to define a research protocol so that replication by other researchers is possible. For this review, the research protocol consists of the following settings: research question, search term, search databases, and inclusion and exclusion criteria of the articles.

Thus, first, the following research questions (RQ1 and RQ2) were defined:

RQ1: How many articles propose the use of fog computing architecture and a real-time solution?

RQ2: How do the authors conceptualize real-time or time constraint when their proposals are based on fog computing architecture?

Based on these questions, a search term was defined that would be able to provide articles that answered RQ1 and RQ2 questions.

("Fog Computing" AND ("Real Time" OR "Time Constraint" OR "Latency Constraint"))

Scopus [20] and Web of Science [21] databases were chosen for the search. The motivation for choosing these bases was because they are widely accepted by the academic community and have a large number of journals and conferences indexed in their base.

Some inclusion and exclusion criteria were adopted in order to restrict the number of articles and select articles strongly associated with research questions:

| **Inclusion Criteria** | **Exclusion Criteria** |
|---|---|
| - Written in English | - In Press Articles |
| - All (without data constraint) | - Conference Review |
| - Areas: Computer Science and Engineering | - Surveys |
| - Type: Conference Proceedings, Journal Article, Book Chapter | - Patents |
| | - Short Articles |
| - Resource: Conferences, Journals, Book Series, and Books | - Close Access Articles |

The next section details the procedures performed to conduct the review, based on the protocol described above.

### 2.2.2. Conducting the Review

After formulated and evaluated, the search protocol was applied to respect three steps: search the databases, first review of articles, and full reading of articles. Figure 2 presents the flowchart of the three steps: search in the database, first review of articles, and full reading of articles.

Searches on databases (Scopus and Web of Science) were performed using the search term presented above.

The Scopus database raised one thousand seventy four (1074) articles, published from 2012. The first refinement was performed—in other words, the application of the inclusion and exclusion criteria of the articles, which resulted in 551 articles.

On the other hand, the search in the Web of Science database resulted in 656 articles, published since 2013. After the first refinement, 276 articles were obtained.

Searches performed in both databases generated a total of one thousand seven hundred and thirty (1730) articles. Applying the inclusion and exclusion criteria in each database resulted in a total of 827 articles. At the end of this stage, the first analysis of the articles was started.

The first review of the articles step aims to perform the first selection of articles, considering the significant amount of articles obtained by the databases.

Thus, repeated articles, posters, extended abstracts, and short articles were removed, since this type of article presents little detail of the work. Reviews, works that proposed a survey of the concepts, use and challenges of fog computing, and the articles with closed access have also been removed.

The titles and abstracts were read; with that, we removed the works that did not propose the use of fog computing but cloud computing. The articles that conceptualized edge computing and fog computing as the same approach were considered. The completion of this step resulted in 257 articles, which were read and analyzed in detail.

Finally, the full reading of articles step consists of reading and thorough analysis of the 257 articles. We selected articles that presented the use of fog and/or edge computing and real-time, time constraint or delay constraint. In total, 95 articles were selected whose results of the analysis are presented in the next subsection.

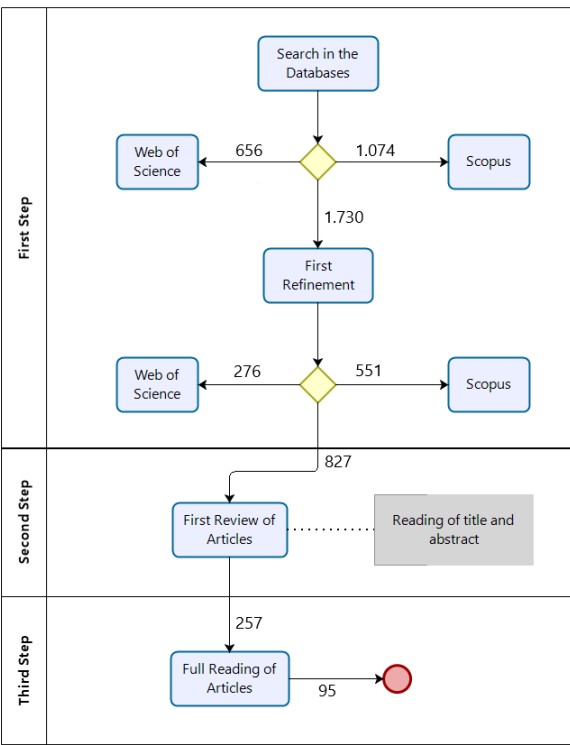

**Figure 2.** Search protocol fluxogram.

### 2.2.3. Analysis of Results

The analysis of results can be divided into two stages: quantitative analysis and classification of articles according to similar characteristics. In the first stage, we present some metrics based on the characteristics of the articles as follows: number of articles published over of the years, country of the first author, and number of articles published by publishers.

We can see in Figure 3 that publications began in 2015 and had a gradual increase, having its apex in 2018 with 33 articles published. Countries such as China (20), the United States (10), and India (9) had the highest number of publications, as shown in Figure 4. For this analysis, we consider the country of the first author of the article.

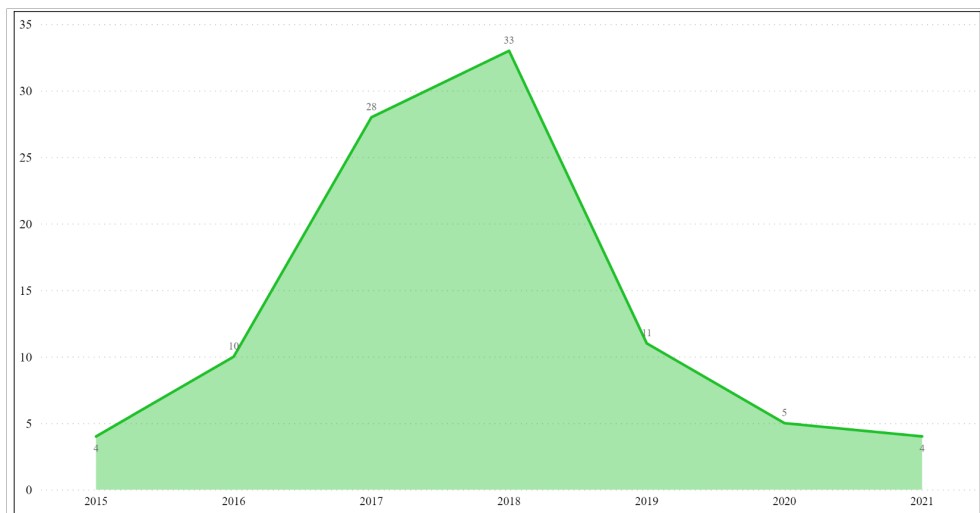

**Figure 3.** Publications separated by year.

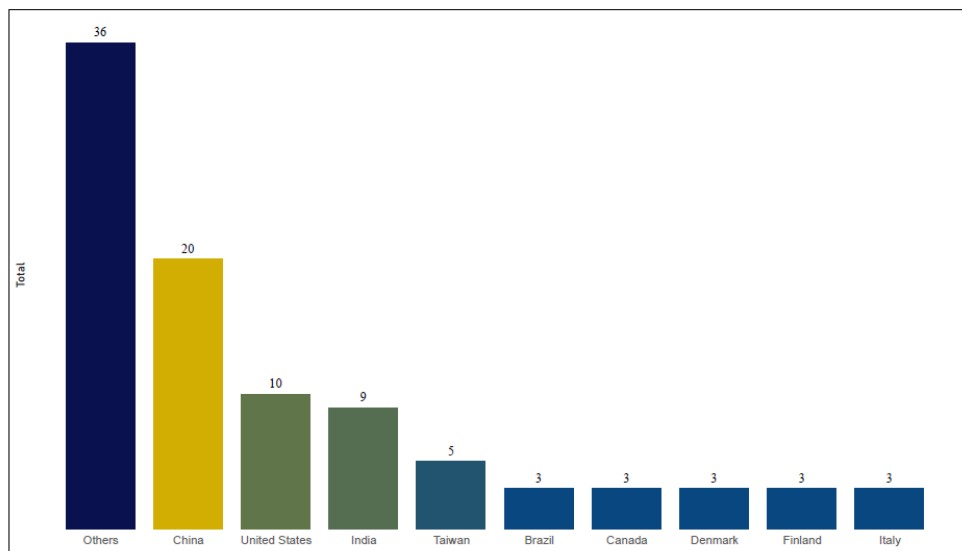

**Figure 4.** Publications separated by country.

Regarding publishers, we separate articles according to the years and types of articles that publishers have most published. As we can see in Figure 5, IEEE had a higher amount of publication every year compared to other publishers. However, it obtained the highest number of publications in 2017 (23), followed by ACM, with four publications in 2018 and Elsevier with four in 2019. In 2019, Elsevier showed an increase in the number of publications compared to IEEE.

Table 1 presents the number of publications by type and publisher, considering articles published in the journal (article) and articles published in event annals (inproceedings). As we can see, there are more congress publications, but it is not a significant difference since inproceeding articles have 53% of journal publications and 47%.

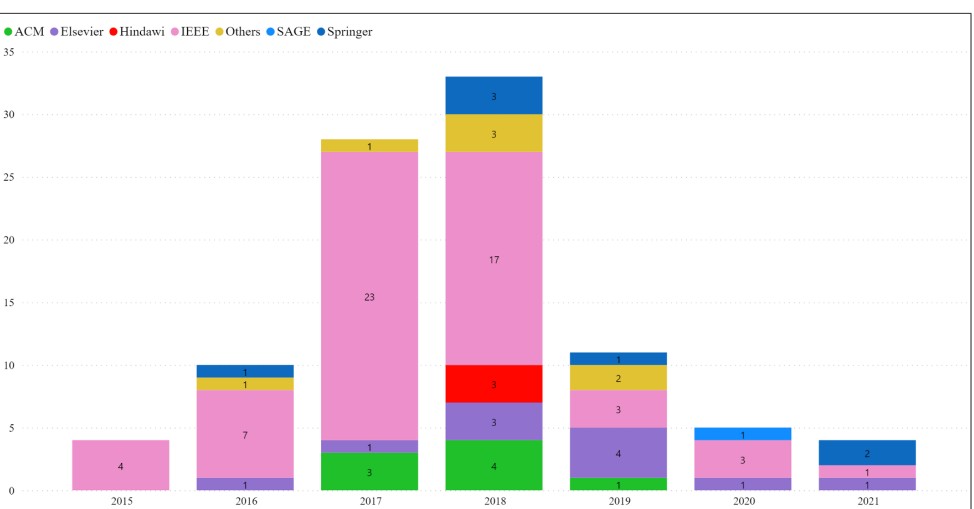

**Figure 5.** Publications by year and publisher.

**Table 1.** Publications by type and publisher.

| Publisher | Article | Inproceeding | Total |
|-----------|---------|--------------|-------|
| IEEE | 19 | 39 | 58 |
| Elsevier | 11 | - | 11 |
| ACM | - | 8 | 8 |
| Others | 6 | 2 | 8 |
| Springer | 6 | 1 | 7 |
| Hindawi | 3 | - | 3 |
| Total | 45 | 50 | 95 |

With the reading of the articles, we have identified a pattern in the way real-time concepts are applied to research. With the identification of this pattern, we classified the articles according to the proposal and the application of the real-time concept in the research. With that, we carried out the second stage of the analysis of the results. The proposed classification is divided into five categories, which are:

- Fog Computing Concept: articles belonging to this category associate real-time to the concept of fog computing. When Bonomi et al. [3] presented fog computing, they attributed, as one of its main features, the possibility of the architecture to execute applications in real-time. In other words, these articles did not present in their proposals' time constraints on application execution or data presentation.
- Low Latency: articles in this category associate low latency with real-time systems. In other words, these articles propose solutions that improve latency and response time rather than time-constrained proposals. Latency, in turn, is the system delay in processing the request.
- Fast Response: articles in this category present understanding that a real-time system responds quickly or instantly. The authors do not have any system response time or deadline constraints, only the improved average response time. Response time, in turn, is the time that the system takes to perform the processing and send the response.
- Data Streaming Applications: articles belonging to this category relate the concept of real-time with data streaming applications. Due to characteristics such as speed and continuous data sending, the authors understand that this type of application performs its tasks in real-time.
- Time, Latency, or Delay Constraint: in articles belonging to this category, the authors proposed time, delay, or latency constraint and understand that the proposed solutions should execute within a time limit or meet an acceptable delay rate.

From the classification of articles by categories, we performed analyses for information such as publication year, country's first author, publishers, and the applications used in the proposals.

Table 2 shows the number of articles published by publisher and category. As we can see, the "Fog Computing Concept" category has the largest number of articles (38), followed by the "Faster Responds" (20) and "Time, Delay or Latency Constraint" (16) categories. Still in Table 2, we can analyze that the publishers IEEE, Elsevier, and Springer obtained more publications in the category "Fog Computing Concept", with 20, 5, and 5 articles published, respectively.

**Table 2.** Publications separated by publisher and category.

| Publishers | Data Streaming Application | Faster Response | Fog Computing Concept | Low Latency | Time, Delay or Latency Constraint | Total |
|---|---|---|---|---|---|---|
| IEEE | 4 | 14 | 20 | 9 | 11 | 58 |
| Elsevier | 1 | 3 | 5 | 2 | - | 11 |
| ACM | 2 | - | 4 | - | 2 | 8 |
| Springer | 1 | - | 5 | - | 1 | 7 |
| Hindawi | - | 2 | - | - | 1 | 3 |
| MDPI | - | 1 | - | - | 1 | 2 |
| Chulalongkorn University | - | - | - | 1 | - | 1 |
| CLOSER | - | - | - | 1 | - | 1 |
| IOP | - | - | 1 | - | - | 1 |
| SERSC | - | - | 1 | - | - | 1 |
| Taylor & Francis | - | - | 1 | - | - | 1 |
| SAGE | - | - | 1 | - | - | 1 |
| Total | 8 | 20 | 38 | 13 | 16 | 95 |

The number of publications per year and category is shown in Figure 6. We can see that the articles belonging to the categories "Fog Computing Concept", "Low Latency", "Time, Delay or Latency Constraint", and "Data Streaming Application" had more publications in the year 2018 with 13, 6, 6, and 3 articles, respectively. On the other hand, the category "Faster Response" had more publications in 2017 (9 articles).

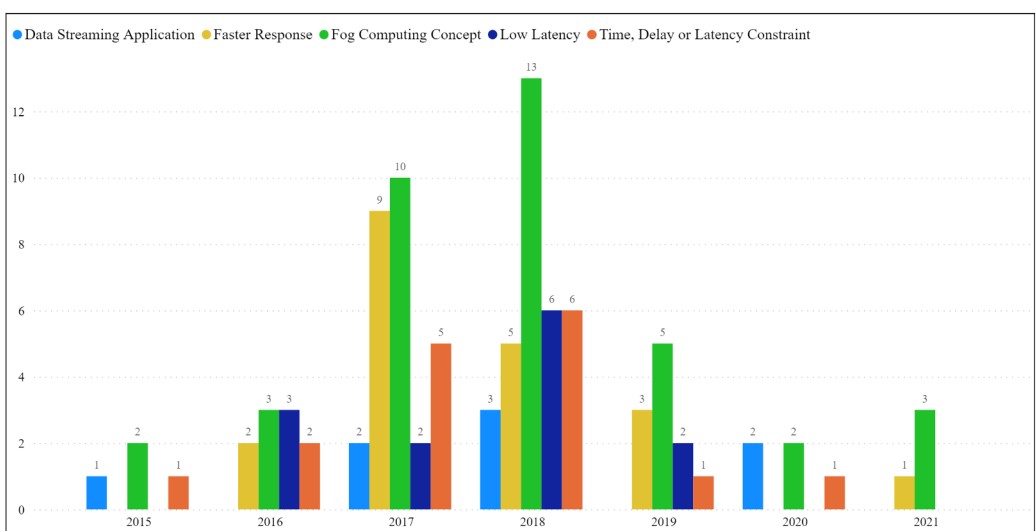

**Figure 6.** Publications separated by year and category.

Finally, we analyze the applications used by the authors to implement their proposals. Table 3 shows the number of published articles separated by categories and applications. It can be analyzed that the applications "Health" (18), "Smart Vehicular" (17), and "Smart City" (13) have the largest number of publications in the "Fog Computing Concept", "Faster Response", and "Low Latency" categories, respectively.

**Table 3.** Publications separated by application and category.

| Application Type | Data Streaming Application | Faster Response | Fog Computing Concept | Low Latency | Time, Delay or Latency Constraint | Total |
|---|---|---|---|---|---|---|
| Health | 1 | 2 | 13 | 2 | - | 18 |
| Smart Vehicular | 2 | 5 | 5 | 3 | 2 | 17 |
| Smart City | 2 | 3 | 6 | 2 | - | 13 |
| Scheduling Algorithm | - | - | - | - | 6 | 6 |
| Smart Home | - | - | 3 | 1 | - | 4 |
| Smart Things | - | 2 | 1 | - | 1 | 4 |
| Task Scheduling | - | - | - | - | 3 | 3 |
| Algorithm | - | - | - | 1 | 1 | 2 |
| Data Analytics | 2 | 1 | - | - | - | 3 |
| Face Recognition | - | - | 1 | 1 | - | 2 |
| Security | - | 1 | 1 | - | - | 2 |
| Smart Parking | - | 2 | - | - | - | 2 |
| Asset Tracking | - | - | 1 | - | - | 1 |
| Device-to-Device | - | - | - | 1 | - | 1 |
| Energy Save | - | 1 | - | - | - | 1 |
| Network | - | 1 | - | - | - | 1 |
| Protocol | - | - | - | 1 | - | 1 |
| Scheduling | - | - | - | - | 1 | 1 |
| Signal Anomalies | - | 1 | - | - | - | 1 |
| Smart Grid | - | - | 1 | - | - | 1 |
| Virtualization | 1 | - | - | - | - | 1 |
| IoT Applications | - | 1 | - | - | $-1$ | - |
| Total | 8 | 20 | 38 | 13 | 16 | 95 |

The next section details the articles belonging to each of the categories.

### 2.3. Considerations

We begin this section with the material, which consists of a description of the terms and approaches used in this article, that is, Edge Computing, Fog Computing, Real-time, and Near Real-time. We verified divergences in the concept of edge and fog computing. There is a group of researchers who understand that, in an IoT architecture, processing and communication are performed at the edge layer and, after that, the data are sent to the cloud for analysis. A second group considers the edge computing as a layer of devices with some processing and communication capacity. However, the edge layer sends the data to an intermediate layer, called fog computing, for the processing and presentation of the data, which will later send the results to the cloud computing for persistent storage. Fog computing, in turn, is conceptualized as an intermediate layer, between the edge and cloud computing layer that receives, transforms, processes, and presents the data to the user. Therefore, in this article, the authors consider that the IoT architecture is composed of the edge, fog, and cloud computing layers. In other words, we do not consider that the data processing is performed at the edge layer, and we consider that there is a need to have an intermediate layer to perform such role.

In addition, we present three ways of conceptualizing real-time systems and we could see that there is also divergence in this conceptualization. Systems that make the best effort to perform their tasks are known as systems that respond near real-time. In view of this divergence of conceptualization of the real-time approach, we conducted an extensive search for articles in order to search for different concepts used to define real-time, which consists of the research methods used to develop our article.

The research method consists of planning, conducting, and analyzing the selected articles. A research protocol was carried out with the objective of enabling the replication

of the systematic review. After validating the protocol, a search was performed that returns 95 articles. These articles presented a pattern in the way real-time was defined, which allowed us to create a classification. This classification has five categories, as presented in this section. A brief description of the proposals for the selected articles is presented in the next section.

### 3. Classification of Articles

This section presents a descriptive analysis of articles belonging to each category, detailing their proposals and applications.

#### 3.1. Fog Computing Concepts

Of the 95 selected articles, 38 belong to this classification, as their proposal is a fog computing approach that uses real-time.

Al-Hasnawi and Lilien [22] propose an approach to reducing IoT data privacy rich to protect sensitive data throughout its lifecycle. In other words, this research proposes software called Policy Enforcement Fog Module (PEFM), designed to run on a single fog node. PEFM enforces privacy policies in two different ways: directly on real-time applications executing on local fog nodes, and indirectly on non-real-time applications executing on remote fog or cloud.

The research of Alemneh et al. [23] proposes an architecture that uses pedestrian smartphone sensors to obtain data and detect its geographical location and fog nodes. The architecture, called PV-Alert (Pedestrian-Vehicle Alert), aims to use sensor data to predict collisions between vehicles and pedestrians and send alerts if necessary.

Amadeo et al. [24] propose a Cloud of Things (CoT) platform, called ICN-iSapiens, whose goal is to solve smart home challenges through the use of Information Centric Networking (ICN) and Fog Computing concepts. The proposed architecture is composed of three layers: the physical layer, fog layer, and remote cloud layer. The middle tier (fog) consists of smart homes' servers (HSs) to support real-time services and to abstract the heterogeneity of IoT devices.

Bhargava et al. [25]'s research presents a low-cost Wireless Sensor Networks (WSN)-based system for monitoring real-time mobility and outdoor positioning of older adults with Alzheimer's. The purpose of monitoring is to detect anomalous behaviors and decrease the risk of the elderly to wander. Real-time data analysis is performed on the wearable device itself using the Fog Computing approach.

Cao et al. [26] propose an algorithm and a mobile real-time fall detection system called UFall. The proposed system uses low-cost pervasive smartphones and employs the fog computing paradigm for real-time detection. It has been implemented to monitor patients with a history of stroke due to the high incidence of falls.

Chen et al. [27] propose an intelligent urban traffic surveillance architecture and a single destination multi-vehicle tracking algorithm to reduce complexity and achieve real-time performance. The fog computing-based prototype was designed with four fog nodes and two drones acting as monitoring cameras.

The implementation of the concept of fog computing for an e-Health scenario is proposed in Craciunescu et al. [28]'s article. For this, we propose a real-time signal processing algorithm, implemented in fog nodes, near the detection environment. These nodes are responsible for all health-related real-time data processing to enable a set of personalized services.

Darwish and Abu Bakar [29] propose a big data analysis architecture for Real-Time Intelligent Transportation System (RITS-BDA). The architecture utilizes the facilities of the fog computing approach to enable real-time intelligent transport system in the context of the Internet of Vehicles. The RITS-BDA architecture project considers three perspectives: real-time big data analysis, Internet of Vehicles, and computer intelligence (fog and cloud computing).

Giordano et al. [30] propose an architecture, called Rainbow, that allows the development of applications in smart cities. Rainbow provides the combination of multi-agent systems and fog computing to enable the development of intelligent distributed applications that can interact directly and in real-time with the physical world.

Islam and Hashem [31] propose a data storage and processing architecture called IF3C (Internet of Things Layer; Fog Data Collection Layer; Fog Edge Layer; Fog Intermediate Computation Layer; Cloud Computing). The purpose of the proposed architecture is to provide real-time services from local storage and Edge Data Centers using fog computing infrastructure. In addition, the Ford–Fulkerson and priority-based queue algorithm is used for load balancing and job scheduling.

Hussein et al. [32] proposes an SDN VANET architecture in a 5G environment, to provide easy management and security based on the SDN security plan. In addition, an SDN VANET real-time attack detection and prevention technique with tracking capabilities is proposed.

Liu et al. [33] propose a framework for the hybrid privacy-preserving clinical decision support system called HPCS (Hybrid Privacy-preserving Clinical decision support System) in fog and cloud computing. HPCS uses the fog computing approach to monitor the health condition of patients in real-time reliably.

Mostafa and Mohammad [34]'s article proposes to combine the capabilities of cognitive and context-sensitive fog computing and IoT to develop a cognitive management framework. The goal is, in a context-sensitive manner, to maximize the number of functions that can be performed on the fog, to determine which functions should be performed on the fog and which ones to perform on the cloud, and how the fog should interact with the cloud. The proposed framework must respond in real-time to support a large number of low latency devices, sensors, and applications.

Nandyala and Kim [35] propose an IoT-based real-time u-healthcare monitoring architecture. The proposed architecture makes use of the advantages of fog computing to take advantage of the proximity between processing and the devices and thus provide real-time monitoring for smart homes and hospitals.

Rahman et al. [36] propose a mobile computing framework that supports real-time, location aware custom services for crowds. The framework consists of cloud, fog, and edge computing infrastructure. Each fog node covers a geographic zone and offers a subset of a city's services and features based on a user's geographic location.

Rauniyar et al. [37] propose a crowdsourcing-based disaster management model using the fog computing and IoT model, called Crowdsourcing-based Disaster Management using Fog Computing (CDMFC). In addition, a data transfer mechanism is proposed for the model so that disaster-related IoT data are sent to the fog even if the direct link is not available. The objective of the CDMFC is to detect disasters in real-time and to disseminate information in advance to ensure public safety.

Taneja et al. [38]'s research proposes an assisted application system based on the fog computing approach for animal behavior analysis and health monitoring in a dairy production scenario. The goal of the proposal is to identify farm activities that require a response and decision support in real-time or near real-time.

Tseng et al. [39] propose a real-time automatic scaling mechanism for industrial applications. The proposed mechanism is called Fuzzy-based Real-time Auto-Scaling (FRAS). FRAS aims to be able to be employed in industries with IoT applications to provide a lightweight, automatic fog-based engine for industrial applications.

Verma and Sood [40] propose remote monitoring of the health of smart home patients through the use of the concept of fog computing. The model uses advanced techniques and services such as embedded data mining, distributed storage, and notification services for network devices. In addition, it uses event-based data transmission methodology to process patient data in real-time at the fog layer.

In Yaseen et al. [41]'s research work, a model is proposed to detect collusion attacks in an IoT environment. The proposed model is based on the fog computing approach for

real-time monitoring of possible attacks. In addition, a software-defined system layer is used to add flexibility to the configuration of fog nodes, allowing them to detect various types of collusion attacks.

Zhang and Li [42] propose a data acquisition mechanism based on the fog computing architecture capable of filtering abnormal data and meeting real-time requirements. The proposed mechanism uses a cooperation method through the use of architectural and algorithmic approaches.

Xiao et al. [43] propose a prediction system architecture and condition management of hydropower equipment based on the concepts of fog computing and Docker container. The goal of using the fog computing approach is to improve real-time processing capacity and address network congestion and delay issues.

Dehnavi et al. [44] propose a resource provisioning scheme for partitioning a particular workload across multiple layers of computing (cloud data center, private cloud, fog nodes, and edge nodes), subject to reliability and real-time requirements. The purpose of workload partitioning is to provide design decisions, such as computing resource specification, minimum bandwidth, and the number of replicas required for each application.

Huertas Celdrán et al. [45] propose a concept of the virtual medical device deployed at the edge of the network to facilitate and provide dynamism and low latency to achieve security in Medical Cyber-Physical Systems (MCPS). In addition, a fog computing-oriented framework is proposed to enable dynamic and real-time management of physical and virtual medical devices, as well as the network that makes up MCPS.

In Kaur and Sood [46]'s paper, a three-layer framework based on IoT and fog computing for forest fire detection and real-time fire susceptibility analysis is proposed. For this, sensors are implemented to continuously monitor the environment in order to look for meteorological attributes and phenomena that cause forest fires. In addition, the fog computing layer is used to analyze in real-time the delay-sensitive data captured by the sensors.

Hellmund et al. [47] propose an application that uses the concept of fog computing for real-time facial recognition with smart glasses. Therefore, the Intelligent People Recognition Assistant (IPRA) is proposed, which aims to achieve low latency and power consumption.

Sood and Mahajan [48] propose an IoT-based healthcare and fog computing framework to identify and control early-stage hypertension attacks. The system provides real-time notifications to users and doctors in case of emergency. In addition, it identifies the stage of hypertension based on the data collected by the sensors and predicts the risk level of hypertension attack on local and remote users.

Devarajan et al. [49]'s article proposes an energy-efficient fog computing-based healthcare system to monitor and maintain blood glucose levels in real-time. The J48Graft decision tree is used to predict the level of diabetes risk with the highest classification accuracy. The goal is to monitor physiological conditions and contextual information.

Nguyen Gia et al. [50] propose a system for continuous remote monitoring of IoT-based healthcare and fog computing. The objective of the proposed system is to improve disease analysis and diagnosis accuracy with health (blood glucose, ECG, temperature) and context (ambient temperature, humidity, and air quality) data. An encryption algorithm is applied to the system to protect the collected data. The data are then encrypted on the sensor nodes before being transmitted and decrypted on smart gateways.

Pešić et al. [51] propose a micro-location asset tracking system using the Bluetooth Low Energy (BLE) protocol. The objective of the system is to perform real-time position estimates and asset tracking based on BLE beacons and scanners. To this end, a context-sensitive fog computing system was developed, which includes IoT controllers, sensors, and a cloud computing platform.

Popović and Rakić [52] propose a framework for integrating fog computing-based IoT control systems. The purpose of the proposed framework is to perform context-aware real-time service applicable to time-aware applications.

Ungurean [53] propose a fog node architecture that enables the IoT of devices that make up the industrial environment, integrating a set of fieldbuses such as EtherCAT, Profinet, Modbus TCP/IP, Modbus RTU, CANOpen, and Profibus. An implementation of the Distribution Service for Real-Time System (DDS) protocol is used to interconnect fog nodes. The fog node has the function of meeting the requirements of real-time, data security and integrity, latency, and response time to external events.

Vilela et al. [54] proposes a health monitoring, evaluation, and performance demonstration system based on the fog computing approach. The goal of the system is to minimize data traffic at the core of the network, improve information security, and provide quality information about the patient's health status.

Zhang et al. [55] propose to formulate the task offloading problem as a Multi-Stage Stochastic Programming (MSSP). The goal is to minimize total offload latency, how much workload to offload, how much computing resources to allocate, as well as whether to migrate. The proposed MSSP examines joint offload, resource allocation, and migration decisions, promoting an understanding of the interactions between these decisions.

Hameed et al. [56] propose a load balancing technique that distributes IoT tasks across vehicle clusters. For this, a dynamic clustering approach that considers the position, speed, and direction of vehicles is designed. Additionally, the technique predicts the future position of vehicles considering the dynamic nature of the vehicle network. Simulated results showed a reduction in network delay.

El-Hosseini et al. [57] article presents an IoT-based fire detection model with energy recognition with different multifunctional sensors in smart cities. In the proposed model, a sleep scheduling approach is used to save energy from the sensors. Furthermore, fog computing is applied to process real-time data from a large number of sensors to running systems. The validation of the proposed model was performed through simulation and experimental implementation in a test bench with Arduino, sensors, and Raspberry Pi.

Xu and Zhu [58] propose a computing model based on Lie Group Machine Learning to find possible defective products in production. The model is based on fog computing concept to provide distributed processing.

Wang et al. [59] propose an architecture to combine Limited Memory Eigenvector Recursive Principal Component Analysis (LMERPCA)-based Operational Modal Analysis (OMA) approach and fog computing.

### 3.2. Low Latency

Of the 95 selected articles, 13 belong to the low latency category. This category includes articles that conceptualize real-time systems as being systems that provide low latency.

Batres et al. [60] propose an architecture of communication between vehicles. The implementation uses the Dedicated Short Range Communication (DSRC) as wireless technology and fog computing framework as the basis for the whole architecture. In addition, it utilizes the Wireless Power Transfer (WPT) system for real-time data transfer.

bin Baharudin et al. [61]'s article proposes a system called Cloud/Crowd System that integrates large-scale aggregated and retrieved data into the fog layer. Fog architecture is used since it offers low latency in data processing and storage and allows for geographic location recognition and real-time operations. The purpose of this system is environmental monitoring for early warning system for disaster prevention.

In Feng et al. [62]'s article, a real-time notification protocol called RT-Notification is proposed for applications requiring wireless control. RT-Notification provides low latency TDMA communication between an access point in the fog and portable monitoring devices.

Gia et al. [63] propose a mobility support approach to Wi-Fi based real-time health monitoring systems through an efficient handover mechanism. The handover mechanism helps maintain the connection between the sensor nodes and the low latency system. The proposed approach allows objects and people to be monitored remotely in real-time without any interruption during mobility.

Huang and Xu [64] propose a distributed approach for mobile devices to reliably, and real-time transmit data into the vehicle network through cloud-fog computing. The purpose of the approach is to enable QoS in real-time transmission and impartiality of resource reservation between mobile devices.

Nahri et al. [65] propose an architecture based on three layers: IoV, fog computing, and cloud computing. For the fog computing layer, a framework has been developed to collect and process real-time events generated by intelligent vehicles, as well as view the state of vehicle traffic at each stretch of the road.

Singh et al. [66]'s article provides the details of the real-time implementation of a Deviceto-Device (D2D) based Mobile Edge Fog Computing system. The purpose of the proposal is to explore the benefits of the D2D model by incorporating D2D functionality into the proposed relay gateways.

Wu et al. [67] propose a cyber-manufacturing system based on the fog computing approach. The goal of the proposed system is to provide real-time remote sensing, monitoring and scalable, high-performance computing for prognostics and diagnostics. In addition to the fog infrastructure, the system uses wireless sensor networks, cloud computing, and machine learning.

Yacchirema et al. [68]'s research proposes an architecture of an obstructive sleep apnea monitoring system based on IoT and big data approaches. The proposed three-tier architecture integrates the characteristics of fog and cloud computing to provide sleep apnea diagnosis and treatment. For this, a variety of services, namely remote monitoring, real-time alert notifications, data analysis, and information visualization, are included.

Boulkaboul et al. [69] propose a data management platform to monitor and control web/mobile IoT layer applications using IPv4/IPv6 and higher layer protocols such as CoAP, HTTP, and WebSocket. The proposed platform allows the detection of anomalies in IoT devices and real-time error reporting mechanisms.

Jia et al. [70] propose a smart street lamp (SSL) mechanism based on the concept of fog computing for smart cities. The proposed SSL consists of three main parts: an intelligent sensor street lamp, which can adjust the lamp brightness autonomously; a network used for real-time communication between the manager and the lampposts; and a management platform to notify of lamp breakage and automatically send an employee to service.

Nguyen et al. [71] propose a three-tier architecture consisting of IoT, edge computing, and cloud computing to integrate edge and cloud infrastructures to support IoT applications. In addition, a real-time data transport (RDTS) scheduling scheme for transmitting data from IoT devices to cloud systems is proposed. The RDTs allocate bandwidth resources in layers of IoT and edge by assigning weights to the network capacity can be distributed to all devices.

Finally, the article [72] proposes a model for age-to-age facial recognition with MobileNets to determine a user's identity in the age-changing appearance situation based on the fog computing architecture. The proposed approach incorporates the concept of cooperative work in a distributed system to respond quickly to real-time user authentication requests to meet mobile needs.

### 3.3. Fast Response

Fast response is the second category with the largest number of articles. It contains 20 of the 95 selected articles. As presented, in this category, the authors conceptualize real-time as a fast response. The articles belonging to this category are described below.

Abdulkader et al. [73]'s research aims to design a secure model for intelligent parking space management, control, and monitoring based on the integration between Wireless Sensor Network (WSN), Radio Frequency Identification (RFID), Ad Hoc, and Internet of Things (IoT). The proposed model provides real-time information to detect and reserve parking spaces to reduce traffic and optimize management.

Ali and Ghazal [74] propose a Real-time Heart Attack Mobile Detection Service, called RHAMDS, that utilizes IoT and network devices to prevent vehicular collisions and

improve emergency response time. The authors propose using e-Health IoT devices and Software Defined Network (SDN) in a VANET architecture for reliable performance.

Anderson et al. [75] propose a Real-Time Car Parking System, with support for Edge-Fog infrastructure and Vehicular cloud computing. The purpose of the proposal is to reduce congestion by presenting available parking spaces, improve parking management, and provide vehicle safety in the home environment.

In Brennand et al. [76]'s research, a real-time route management system, called FOg RoutE VEhiculaR (FOREVER), is proposed. FOREVER is an architecture based on the fog computing approach, as the information is close to the monitored area, with communication becoming more efficient and the system response time faster. The purpose of the proposal is to allow vehicle traffic congestion to be reduced by reducing travel time and thereby reducing fuel consumption.

Clemente et al. [77] propose fog computing middleware for real-time Distributed Co-operative Data Analytics (DCDA). The proposed middleware enables processing, analysis, and sharing of information in high-level situations. Test results have shown that DCDA middleware enables scalable and fault-tolerant data analysis and is suitable for real-time situational awareness even with bandwidth and communication constraints.

Costa et al. [78] introduce the concept of Smart Cargo that deals with the complexity of multimodal freight transport holistically. It is a system capable of responding to real-time situations, finding and understanding alternatives, communicating synchronously or asynchronously with other transport modules and loads, and computing adaptive behaviors to optimize their own decisions.

Nguyen Gia et al. [79] propose an IoT-based system for low-cost remote health monitoring, with fog computing and energy-efficient wearable sensors. The proposed system is capable of obtaining vital sign data and ambient temperature and humidity data, and transmitting this data wirelessly for real-time notification and monitoring.

Huo et al. [80]'s article proposes a framework designed for the fog computing architecture for real-time data aggregation with adaptive differential event privacy (Re-ADP). Based on event adaptive privacy, the framework can protect all data collected by sensors.

Raafat et al. [81] propose techniques for detecting signal abnormalities in an IoT system and extracting important data from sensors in real-time. For this, homoscedasticity techniques and statistical resources are introduced. The proposed system can be applied in multiple IoT applications based on sensor data analysis such as motion detection, fire, or occupancy.

Samaniego and Deters [82]'s article proposes Smart Things, software that performs real-time autodetection and self-monitoring tasks. In addition, an expert C Language Integrated Production System (CLIPS) system for performing data analysis and self-deduction, as well as a decentralized fog architecture that integrates permission-based blockchain protocols, is proposed in this research.

Sinaeepourfard et al. [83]'s article proposes a fog-to-cloud data management architecture for smart cities with the data preservation block approach. The proposed architecture utilizes the benefits offered by fog and cloud computing technologies, and provides real-time data access through the data preservation model, faster than in a centralized architecture.

Sharma and Wang [84] propose a framework for collaborative edge-cloud processing to handle live data analysis on wireless IoT networks. The purpose of the proposed framework is to utilize cloud storage and computing capacity and to process delay-sensitive edge computing tasks to ensure real-time processing and feedback.

Soultanopoulos et al. [85] propose an IoT service architecture and gateway system implementation. The purpose of the proposal is to allow the system to run on smart devices (smartphones, tablets) that support access to Bluetooth Low Energy (BLE) compatible with sensors and allow real-time sensor registration through a simple scheme of data usage.

Wang et al. [86]'s research paper proposes a safe and private VANET navigation scheme based on fog computing. The proposed system uses the crowdsourcing approach

to collect real-time traffic data and analyze it to provide navigation services to vehicle drivers. The purpose of the proposal is to calculate the best destination route according to the vehicle location.

Wang et al. [87] propose a computational offloading-based algorithm for real-time traffic management for IoV and fog computing systems. The goal of the proposal is to minimize the average response time of events reported by vehicles.

Luo et al. [88] propose a hierarchical architecture, called multi-cloud to multi-fog, based on the container concept. The purpose of using a container as a resource unit is to reduce the response time of requests. In addition, an energy balancing based task scheduling algorithm is proposed to improve the life of Wireless Sensor Networks (WSN) without increasing the task delay.

Ning et al. [89] propose a three-tier model, based on the concept of vehicular fog computing, to manage city-wide traffic in a distributed manner. In addition, a real-time traffic management scheme is proposed. The purpose of the proposal is to minimize the response delay for the traffic management system by balancing the load between cloud and fog nodes.

Tsaur and Yeh [90] propose a secure notification scheme based on the concept of fog computing to provide real-time response. The combination of IoV and IoT devices such as eye movement tracking devices and heart rate sensors is also proposed to detect driver fitness and distraction in advance.

Nguyen Gia et al. [79] propose a health monitoring system that provides continuous remote ECG monitoring along with automatic analysis and notification. The system consists of energy-efficient sensor nodes and a fog layer. In addition, the system can represent the collected data in useful ways, perform automated decision-making, and provide advanced services such as real-time notifications for immediate attention.

Finally, Zheng et al. [91]'s article proposes a real-time QoE prediction scheme based on the concept of fog computing. The purpose of the proposal is to observe bidirectional video traffic and predict the QoE of the video stream supported by Dynamic Adaptive Streaming over HTTP (DASH).

### 3.4. Data Streaming Applications

For some authors, systems that manipulate data streams are real-time due to the characteristics that these systems must present, which are: fast response, low latency, and continuous data processing. The eight articles belonging to this category are presented below:

Cao et al. [92] propose a mobile edge computing-based platform on which physical devices are deployed on buses, where a descriptive analysis is implemented to analyze real-time data streams.

Jayaraman et al. [93] propose a context-aware real-time data analysis platform for fog computing called CARDAP. It is a flexible and extensible generic platform capable of efficiently deploying distributed Cyber-Physical Social Systems (CPSS) applications such as MCS (Mobile Crowdsensing) that execute on demand.

In the research of Vinueza Naranjo et al. [94], a virtualized network computing architecture based on the container concept is proposed. The proposed architecture operates as middleware and exploits the native capacity of containers to enable dynamic real-time scaling of available computational and network virtualized resources. The main objective is to reduce the total energy consumed per slot.

Neto et al. [95] propose a fog framework for a smart and secure vehicular environment applied to video monitoring fog computing to improve crime detection cost-effectively. The proposed framework, called Fog-FISVER, aims to support real-time, autonomous crime detection for public transport services.

Wang et al. [96] propose a real-time surveillance system architecture based on the edge computing approach. The goal of the proposed architecture is to adjust computing power resiliently and dynamically route data to edge servers suitable for applications. In addition,

resource reallocation and workload balancing scheme in emergencies are presented in the article.

In Xu et al. [97]'s research, an intelligent surveillance approach to vehicle tracking is proposed. The goal is to explore an intelligent surveillance system to track all vehicles in real-time. Since vehicle tracking is a continuous and geographically distributed task, fog computing infrastructure was used to store data on fog nodes near cameras.

Li [98] propose a smart and real-time healthcare information processing system based on fog computing that processes and analyzes data using Hadoop and Apache Spark systems. Data information collected in the data pre-processing step is further processed and analyzed in real-time using the Hadoop and Apache Spark ecosystem.

Finally, Nguyen et al. [99] propose a multilayer (IoT-device, edge, fog, and cloud) streaming analytics platform with two-tier fog layer, which comprises streaming and analytics tiers. The goal is to be capable of dealing with a large number of streaming and big data analytics IoT-based Cyber-Physical Systems applications. The fog layer uses advanced streaming and analytics paradigms, such as Apache Kafka, Apache Spark, and Hadoop, to support the effective development of applications that use streaming, long-term data storage, and analytics.

### 3.5. Time, Latency, or Delay Constraint

Of the 95 articles, 16 belong to the time, latency, or delay constraint category. In this category, the authors use the concept of real-time as an approach that systems must respond within an acceptable time or delay. The proposals and applications used by the authors are described below.

Barzegaran et al. [100] present the problem of mapping and programming mixed-criticality applications on a fog computing platform. Therefore, a Simulated Annealing meta-heuristic strategy is proposed that uses a simulation of the EDF algorithm to generate the schedules. The purpose of the proposal is to determine the task mapping and scheduling table of their activations to maximize the quality of control of tasks control and to meet time requirements, including non-critical real-time tasks.

Desikan et al. [101] propose a Hungarian-based topology-building algorithm to identify the ideal gateways locations and resources, which consider latency constraints of various applications in a smart city. Each application has its own associated latency constraint, which includes propagation, transmission, processing, and buffering. A candidate location identification algorithm based on computational geometry is also proposed to identify the possible places for the placement of gateways.

A resource scheduling algorithm called Privacy-Aware Scheduling in a Heterogeneous Fog Environment (PASHE) in fog computing environments that support heterogeneous micro data centers (MDCs) and user mobility is proposed in the Fizza et al. [102] article. The proposed approach considers task requirements, such as security and completion constraints, to identify the most appropriate place for execution. Tasks were classified into three categories according to deadlines and privacy of information: private with a tight deadline, performed at local MDC; semi-private with a tight deadline, it runs on remote MDCs; public domain with a loose deadline executes in the cloud data center. The proposed algorithm also allows the reservation of bandwidth on remote MDCs, so that it is possible to meet the established deadlines.

Chen et al. [27] propose a task scheduling model for car systems based on the concept of fog computing that transfers device-level scheduling to the server level. The purpose of the proposed model is to reduce the amount of computation required to perform load balancing. Additionally, a load balancing optimization problem is formulated to minimize missed deadlines and total task execution time.

Fan et al. [103] propose a time-sensitive task scheduling mechanism based on fog computing on a layered IoT infrastructure. To this end, the task scheduling problem was formulated for a cloud-fog environment such as the multidimensional backpack, known to be NP-hard. Then, an algorithmic solution based on the ant colony optimization

heuristic (ACO) is proposed. The goal is to maximize the total net profit received by service providers by scheduling and positioning tasks to meet application deadline requirements and resource capacity constraints in fog and cloud computing.

Kochar and Sarkar [104]'s article proposes a framework for sharing and symbiotic consolidation edge computing capabilities for creating a fog computing environment in real-time with high resource utilization. For this, two scheduling algorithms are proposed: Distributed EDF for Fog (DEF), to maximize resource utilization at the network edge, and Profit-aware Distributed EDF for Fog (PDEF) extends the DEF algorithm making task scheduling profitable as it ensures a minimum amount of profit for the service provider and meets task constraints and device computation rates.

Kopetz and Poledna [105] present the time-triggered virtual machine (TTVM) concept that provides a precisely specified virtual interface between a real-time software component and its underlying hardware infrastructure. TTVM's goal is to make software component allocation more flexible to provide ways to implement fault tolerance, evolution, and online validation in a time triggered distributed architecture.

Park and Yoo [106] propose the Reinforcement Learning Data Scheduling (RLDS) algorithm based on reinforcement learning in order to find the optimal scheduling in case the user requests real-time service. The goal of the algorithm is to minimize the number of services that are not transmitted on time for vehicle applications connected to an SDN network and using the fog computing approach.

Raagaard et al. [107] propose a heuristic scheduling algorithm based on TSN (Time-Sensitive Networking) and the fog computing platform. For traffic with a critical time, TSN uses the type of traffic scheduling depending on GCL (Gate-Control Lists) on each output port of a network switch to decide on the transfer of staggered boards. The purpose of the proposed algorithm is to determine GCLs at runtime so that deadlines are met and queue usage is minimized to accommodate non-critical traffic.

In the Zheng et al. [91] article, a mechanism is proposed for continuous and real-time detection of the boundary region of objects in a fog computing environment. For this, interpolation algorithms were applied to estimate the sensory data in specific geographical locations in order to obtain a more precise boundary line. In addition, the heuristic algorithm was applied to generate ideal paths for mobile sensors. The objective of the proposed mechanism is to obtain a precise region of the object boundaries and to balance the energy consumption and time of the mobile sensors.

Xiao et al. [108] propose a VANET architecture that integrates SDN and fog computing to realize flexible network management and low delay services. A fault-tolerant reliability optimization algorithm is also proposed, which considers each case of failures in the processing nodes and optimizes the reliability of the entire system with delay constraints.

Wang et al. [109] propose a time-constrained multitasking scheduling algorithm for a hybrid Mobile Cloud Computing architecture based on the ant colony optimization algorithm (ACO) in order to find the optimal scheduling scheme. In addition, the design of the proposed algorithm considers load balancing and power consumption. The goal is to maximize total profit and constraints, which primarily include completion time, resource consumption, and execution order.

The Suto et al. [110] article proposes an energy-efficient, time-sensitive wireless computing system for IIoT applications. The system aims to minimize system power consumption by respecting an acceptable delay limit for data collection. For this, the proposed system controls server downtime and the number of network connections.

Wang and Li [111] propose an SDI architecture based on the IIoT platform and fog computing. The purpose of the proposed architecture is to enable real-time task processing with high reliability. To this end, a Computing Mode Selection (CMS) module is designed so that it can be select an optimal computing mode for each task, and a method for determining the execution sequence based on the priority of the tasks.

Singh et al. [112] proposes a security-aware real-time scheduling algorithm called RTSANE. The objective of the proposed algorithm is to support the integration between a

microdata center (MDC) and a cloud data center (CDC) in order to balance performance and data security and privacy constraints. The algorithm distributes tasks to local MDCs and CDCs or remote MDCs and CDCs according to the level of privacy and time constraints.

Finally, Louail et al. [113] propose a real-time dynamic task scheduler which considers deadline and frequency constraints as an emergency degree, at the fog level, in order to process the incoming tasks correctly. A fog network is based on auctions in which tasks rejected by a fog can be accepted at another fog.

Collectively, these studies outline a critical role of fog computing. It was possible to see that the fog computing concept is used in a broad spectrum of applications: vehicular computing, intelligent systems inspections, fire detection platforms, healthcare frameworks, and many others scenarios that embrace smart cities. The majority of the works propose a framework/systems/platforms/approaches/methods that use three or more layers. The fog computing layer is always in the middle of these architectures. The scientific community is actively finding new and better solutions to improve the system quality and, consequently, improving people's daily lives.

## 4. Lessons Learned

The concept of real-time has had several biases and has been used according to the approach proposed in the articles. The classical literature initially proposed that real-time systems had to perform their tasks within a deadline, regardless of the velocity of such processing.

The actual Internet of Things scenarios present a continuous data sending and receiving, which creates the requirement for an enhanced processing and response. Fog computing, in turn, provides local processing and, consequently, low latency and fast response.

Therefore, it is increasingly common to find articles that associate the speed of processing with real-time systems, which corroborates the concept of real-time presented by Stonebraker and Zdonik [114]. The Gomes et al. [15] article presents a survey that shows the association of the data stream concept with systems that respond in real-time. In that survey, the articles were classified according to the way real-time was understood by the authors, based on big data and data stream. Big data tools propose real-time data execution by handling streaming data.

With our research, we could see that the fog computing configuration has been widely used, as it provides benefits such as low latency and fast response, due to the processing, storage, and treatment of the data being close to the edge. Moreover, an IoT environment has a need to get a fast response due to the sensitivity of the information generated by the applications. In other words, there is a greater concern in processing the data immediately after its arrival than with the response time constraint and with system predictability, which directs this type of environment to offer near real-time responses.

Our research performed the classification and categorization of the articles since it was possible to verify a pattern regarding the authors' understanding of the concept of systems with temporal requirements. We have divided the articles into five categories in which three of them present similar real-time concepts. In other words, the "Faster Response" and "Low Latency" categories are directly related to the "Fog Computing Concept" category, since the main characteristics of fog computing are to provide local processing and thus low latency and fast response. We decided to split these articles, as the authors accurately presented how their application should answer, as it was the real-time response, even using the fog computing approach.

The category "Data streaming" included articles whose purpose was the use of video monitoring and data streams. The articles that belong to this category corroborate what was presented by Smith [115], since ordering in the input and output of the data are necessary, mainly concerning the video frames.

On the other hand, the articles belonging to the category "Time, Delay or Latency Constraint" propose mostly (68.75%) the implementation of algorithms or task scheduling methods.

Only 31.25% propose the implementation of time constraints for industrial and smart things applications. However, there is concern about the need for guarantees and predictability in the processing and delivery of health and smart vehicular data [116–118]. These are time-sensitive applications in which delay or failure can cause catastrophic consequences.

Table 4 shows the summary list of selected articles, the year of publication, the application used, and the category in which it was inserted.

**Table 4.** List of selected articles.

| Article | Year | Application | Category |
|---------|------|-------------|----------|
| Cao et al. [92] | 2017 | Smart City | |
| Jayaraman et al. [93] | 2015 | Data Analytics | |
| Vinueza Naranjo et al. [94] | 2018 | Virtualization | |
| Neto et al. [95] | 2018 | Smart Vehicular | |
| Wang et al. [109] | 2017 | Smart City | Data Streaming Application |
| Xu et al. [97] | 2018 | Smart Vehicular | |
| Nguyen et al. [99] | 2020 | Data Analytics | |
| Li [98] | 2020 | Health | |
| Abdulkader et al. [73] | 2018 | Smart Parking | |
| Ali and Ghazal [74] | 2017 | Health | |
| Anderson et al. [75] | 2017 | Smart Parking | |
| Brennand et al. [76] | 2017 | Smart City | |
| Clemente et al. [77] | 2017 | Smart City | |
| Costa et al. [78] | 2016 | Smart Vehicular | |
| Nguyen Gia et al. [79] | 2017 | Health | |
| Huo et al. [80] | 2018 | Security | |
| Luo et al. [88] | 2019 | Energy Save | |
| Ning et al. [89] | 2019 | Smart Vehicular | |
| Raafat et al. [81] | 2017 | Signal Anomalies | |
| Samaniego and Deters [82] | 2017 | Smart Things | Faster Response |
| Sharma and Wang [84] | 2017 | Data Analytics | |
| Sinaeepourfard et al. [83] | 2018 | Smart City | |
| Soultanopoulos et al. [85] | 2016 | Smart Things | |
| Tsaur and Yeh [90] | 2019 | Smart Vehicular | |
| Wang et al. [86] | 2017 | Smart Vehicular | |
| Wang et al. [87] | 2018 | Smart Vehicular | |
| Zheng et al. [91] | 2018 | Network | |
| Srirama et al. [119] | 2021 | IoT Applications | |
| Alemneh et al. [23] | 2017 | Smart Vehicular | |
| Al-Hasnawi and Lilien [22] | 2017 | Smart Home | |
| Amadeo et al. [24] | 2017 | Smart Home | |
| Bhargava et al. [25] | 2017 | Health | |
| Cao et al. [26] | 2015 | Health | |
| Huertas Celdrán et al. [45] | 2018 | Health | |
| Chen et al. [27] | 2017 | Smart City | |
| Craciunescu et al. [28] | 2015 | Health | |
| Darwish and Abu Bakar [29] | 2018 | Smart Vehicular | |

**Table 4.** *Cont.*

| Article | Year | Application | Category |
|---|---|---|---|
| Dehnavi et al. [44] | 2019 | Industrial | |
| Devarajan et al. [49] | 2019 | Health | |
| Nguyen Gia et al. [50] | 2019 | Health | |
| Giordano et al. [30] | 2016 | Smart City | |
| Hellmund et al. [47] | 2018 | Face Recognition | |
| Hussein et al. [32] | 2017 | Smart Vehicular | |
| Islam and Hashem [31] | 2018 | Smart Grid | |
| Kaur and Sood [46] | 2019 | Smart City | |
| Liu et al. [33] | 2018 | Health | |
| Mostafa and Mohammad [34] | 2017 | Smart Vehicular | |
| Nandyala and Kim [35] | 2016 | Health | |
| Pešić et al. [51] | 2018 | Asset Tracking | |
| Popović and Rakić [52] | 2018 | Industrial | |
| Rahman et al. [36] | 2017 | Smart City | Fog Computing Concept |
| Rauniyar et al. [37] | 2016 | Smart City | |
| Sood and Mahajan [48] | 2017 | Health | |
| Taneja et al. [38] | 2018 | Health | |
| Tseng et al. [39] | 2018 | Industrial | |
| Ungurean [53] | 2018 | Industrial | |
| Verma and Sood [40] | 2018 | Health | |
| Vilela et al. [54] | 2019 | Health | |
| Xiao et al. [43] | 2018 | Industrial | |
| Yaseen et al. [41] | 2018 | Security | |
| Zhang and Li [42] | 2017 | Smart Home | |
| Zhang et al. [55] | 2021 | Health | |
| El-Hosseini et al. [57] | 2021 | Smart City | |
| Wang et al. [59] | 2020 | Smart Things | |
| Hameed et al. [56] | 2020 | Smart Vehicular | |
| Zhang et al. [55] | 2021 | Industrial | |
| bin Baharudin et al. [61] | 2018 | Smart City | |
| Batres et al. [60] | 2016 | Smart Vehicular | |
| Boulkaboul et al. [69] | 2019 | Smart Home | |
| Feng et al. [62] | 2017 | Protocol | |
| Gia et al. [63] | 2018 | Health | |
| Huang and Xu [64] | 2016 | Smart Vehicular | |
| Jia et al. [70] | 2018 | Smart City | Low Latency |
| Nahri et al. [65] | 2018 | Smart Vehicular | |
| Nguyen et al. [71] | 2019 | Algorithm | |
| Singh et al. [66] | 2016 | Device-to-Device | |
| Wang et al. [72] | 2018 | Face Recognition | |
| Wu et al. [67] | 2017 | Industrial | |
| Yacchirema et al. [68] | 2018 | Health | |
| Barzegaran et al. [100] | 2019 | Task Scheduling | |
| Chen et al. [27] | 2017 | Scheduling | |
| Desikan et al. [101] | 2018 | Algorithm | |
| Fan et al. [103] | 2017 | Task Scheduling | |
| Fizza et al. [102] | 2018 | Scheduling Algorithm | |
| Kochar and Sarkar [104] | 2016 | Task Scheduling | |
| Kopetz and Poledna [105] | 2016 | Smart Vehicular | |
| Park and Yoo [106] | 2018 | Scheduling Algorithm | Time Constraint |
| Raagaard et al. [107] | 2017 | Scheduling Algorithm | |
| Singh et al. [112] | 2017 | Scheduling Algorithm | |
| Suto et al. [110] | 2015 | Industrial | |
| Wang et al. [109] | 2018 | Scheduling Algorithm | |
| Wang and Li [111] | 2018 | Industrial | |
| Xiao et al. [108] | 2017 | Smart Vehicular | |
| Zheng et al. [91] | 2018 | Smart Things | |
| Louail et al. [113] | 2020 | Scheduling Algorithm | |

We perform a search, in SCOPUS, for keywords with the following search string: *("Fog Computing" AND ("Near Real Time" OR "Almost Real Time") AND "Real Time")*, and filter to show information from the last six years and only from the engineering and computing areas. The goal of this search was to obtain the trend in the use of temporal concepts in fog computing environments. As a result, we obtained the word cloud shown in Figure 7.

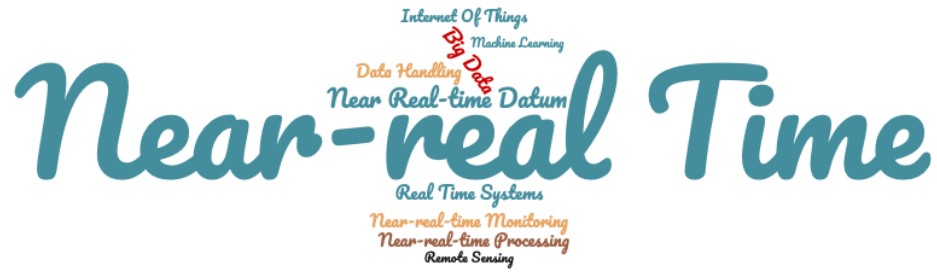

**Figure 7.** Tag cloud from the keywords used in the survey of the publications.

We can analyze that the near real-time concept is being used in approaches as Big data, Internet of Things, and Remote Sensing, which reinforces that these environments tend not to respond in hard real-time, due to the characteristics they present. In addition, it reinforces that, for these environments, there is a greater concern with the fast and continuous processing of data, thus generating a better effort in response time.

Therefore, we intend, as future work, to show that a distributed IoT environment with fog computing configurations offers a better effort to perform its tasks in an acceptable time. In other words, we intend to develop a model and architecture that receives, transforms, processes, stores, and presents data near real-time considering the characteristics of the application.

## 5. Conclusions and Future Works

In this article, we carried out a comprehensive survey focusing on time-sensitive applications implemented in fog computing architectures. The aim of our literature review was to get the state of the art and identify how the concept of real time is being approached by the scientific community and which applications are being studied and implemented.

For this, we propose and execute a research protocol in order to answer two questions: How many articles propose the use of fog computing architecture and a real-time solution? and How do the authors conceptualize real-time or time constraint when their proposals are based on fog architecture? We searched the articles in the Scopus and Web of Science databases, which returned 1730 articles, of which 95 answered our research questions.

We performed an analysis of the selected articles and obtained the number of publications made by year and country, as well as by type of publication (journal or conference) and publishers. In addition, we observed a pattern in the way the concept of real-time was presented by the authors, which allowed us to classify the articles into five categories, which are: "Fog Computing Concept" (38 articles), "Faster Response" (20 articles),"Low Latency" (13 articles), "Data Streaming Application" (8 articles), and "Time, Delay or Latency Constraint" (16 articles). Additionally, we check the type of application implemented in the proposals. The three most used applications were: Health (18 articles), Smart Vehicular (17 articles), and Smart City (13 articles).

As a future work, we intend to design and model an environment with a fog computing configuration so that it presents near real-time response time. In other words, we intend that the model allows the processing of the data to be carried out immediately after its arrival and that the environment presents a best effort so that the response to the processing is within acceptable times and/or delays.

**Author Contributions:** Conceptualization, E.G.; methodology, E.G.; validation, E.G., M.D. and C.D.R.; formal analysis, E.G., M.D. and C.D.R.; data curation, E.G., F.C.; writing—original draft preparation, E.G., M.D., C.D.R.; writing—review and editing, M.D., C.D.R., F.C., P.P., visualization, M.D., C.D.R., F.C., P.P., supervision, M.D.; co-supervision, P.P.; All authors have read and agreed to the published version of the manuscript.

**Funding:** This study was financed in part by the Coordenação de Aperfeiçoamento de Pessoal de Nível Superior–Brasil (CAPES)–Finance Code 001.

**Conflicts of Interest:** The authors declare no conflict of interest.

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
