# Peer review of "A Survey from Real-Time to Near Real-Time Applications in Fog Computing Environments"

_telecom, doi:10.3390/telecom2040028_

Round 1

Reviewer 1 Report

  1. In this paper, the authors present the survey on (near) real-time application in fog computing. The discussion about edge computing and the comparison between edge and fog computing is redundant in Section 2.1.
  2. In Section 3.1, the first paragraph introduces the application about fog computing. Why the reference 34 be quoted? Its disadvantage is the lack of real-time application.
  3. There are 4 categories about the (near) real-time application of fog computing. What is the classification basis? Do they exist a novel category in other references?

Author Response

Dear,

We thank you for your considerations and I would like to inform you that our responses and changes are attached.

Best regards

Reviewer 2 Report

In this work, the authors present a survey about time -sensitive applications running on fog computing systems.  It is a well written paper and well-structured.  

I think it is a good work but, it could be improved. 

Here are my suggestions for the authors:

  • In Table 4, the authors could add one more column to describe the considering issues about the near real-time applications of each work.
  • The authors could be further discussed the implications of findings of discussed papers.

Moreover, I have the following comments:  There are some typos, please check them. For example, in line429, (447, etc.): remove the letter s from the word hands

Author Response

(The authors gave the same response as above.)

Reviewer 3 Report

The article presents an interesting and modern area of researches, the survey is well organized and written. 

Few comments to revise the article:

  • The introduction section is brief, it needs to clearly present the motivation for choosing the survey subjects and more details about each subject
  • In section 3, descriptions of the related works are very brief, usually, the survey readers look for more details about the researches rather than the provided statistics details.
  • both section 4 and 5 titles include future work, while it is just presented in section 5

Author Response

(The authors gave the same response as above.)
